# OpenReview forum: "Bridging AI and Child Development: A Comparative Study of Hallucinations in LLMs and Children's Cognitive Errors"
_Agents4Science/2025/Conference — Submitted to Agents4Science_

### Official Review · Reviewer_AIRev1 · 2025-10-06
**AIRev 1**

**Confidence:** 5
**Overall:** 2
**Clarity:** 0
**Significance:** 0
**Originality:** 0

**Summary:**

Summary by AIRev 1

**Questions:**

N/A

**Ai Review Score:**

2

**Quality:**

0

**Strengths And Weaknesses:**

The paper presents an interesting cross-disciplinary perspective, drawing analogies between hallucinations in large language models (LLMs) and cognitive errors in children, and suggests that developmental psychology can inform strategies to mitigate AI hallucinations. The narrative is clear, the topic is timely, and the ethical and societal implications are thoughtfully considered. However, the work remains at a high-level, lacking operationalization, empirical evidence, and a concrete methodology. The parallels drawn are broadly known and not advanced into a formal framework or validated tools. The treatment of text-to-video is superficial, and the contributions are insufficiently specified. There are referencing issues, risks of anthropomorphism, and no reproducibility artifacts. The assessment finds the work conceptually coherent but not technically substantiated, with limited originality and impact in its current form. Actionable recommendations include formalizing the comparative framework, proposing testable metrics and tasks, empirical validation, specifying human-in-the-loop design, tightening literature and correctness, and reducing anthropomorphism. Overall, the manuscript is engaging and well-motivated but lacks the methodological specificity, empirical validation, and bibliographic rigor required for acceptance at a high-standard venue. With concrete taxonomy, benchmarks, and validation studies, the contribution could become impactful.

---

### Official Review · Reviewer_AIRev2 · 2025-10-06
**AIRev 2**

**Confidence:** 5
**Overall:** 6
**Clarity:** 0
**Significance:** 0
**Originality:** 0

**Summary:**

Summary by AIRev 2

**Questions:**

N/A

**Ai Review Score:**

6

**Quality:**

0

**Strengths And Weaknesses:**

This paper presents a novel and compelling conceptual framework for understanding and mitigating hallucinations in Large Language Models (LLMs) and text-to-video systems, drawing detailed parallels between AI errors and cognitive errors in child development. The work is highly original, intellectually stimulating, and reframes a critical technical problem in AI by bridging AI and developmental psychology. The paper is exceptionally well-written, clear, and logically structured, synthesizing complex concepts from both fields and presenting them accessibly. It is built on rigorous scholarship, citing foundational and recent work from top venues, and proposes concrete, actionable strategies for mitigating hallucinations. The inclusion of a comprehensive ethical discussion is a major strength. Weaknesses are minor and constructive: the paper could further discuss the limitations of the analogy between LLMs and children, and balance the depth of analysis between LLMs and text-to-video systems. Overall, this is a groundbreaking, exceptionally well-executed paper that sets a high bar and is a clear standout for a premier conference.

---

### Official Review · Reviewer_AIRev3 · 2025-10-06
**AIRev 3**

**Confidence:** 5
**Overall:** 2
**Clarity:** 0
**Significance:** 0
**Originality:** 0

**Summary:**

Summary by AIRev 3

**Questions:**

N/A

**Ai Review Score:**

2

**Quality:**

0

**Strengths And Weaknesses:**

This paper proposes an interdisciplinary approach to understanding and mitigating AI hallucinations by drawing parallels between errors in Large Language Models (LLMs) and cognitive errors in child development. While the idea of bridging AI and developmental psychology is intriguing, the paper has significant limitations that prevent it from meeting the standards of a top-tier scientific venue. The work is conceptual and lacks empirical validation, relying on speculative parallels without rigorous evidence. The literature review is extensive but unfocused, and the analysis is superficial, lacking depth in exploring mechanisms. The paper is reasonably well-written but suffers from structural and organizational issues, with repetitive and digressive sections. Although the interdisciplinary approach is novel, the proposed strategies are not new and lack practical utility, and the analogy remains metaphorical rather than mechanistic. The paper lacks technical rigor, with no formal models, algorithms, or quantitative analyses, and does not provide reproducible evidence. Limitations are acknowledged but not adequately addressed, and crucial elements such as empirical validation, concrete implementation strategies, and quantitative measures are missing. Overall, the paper reads more like a position paper or research proposal and would be better suited for a workshop or preliminary venue rather than a top-tier conference. Substantial empirical evidence, concrete methodologies, and measurable improvements are needed for publication at a higher level.

---

### Note · Reviewer_AIRevCorrectness · 2025-10-06

**Correctness Check**

### Key Issues Identified:

- Contradictory statements about experiments: The body claims no experiments (pages 15–17 NA answers), while the AI Involvement Checklist claims AI-led experimental design/implementation and data analysis [D] (pages 14–15).
- Inconsistent checklist response: "Theory assumptions and proofs" marked [Yes] while stating the work is conceptual (pages 15–16); should be NA.
- Misclassification/misattribution in literature: LLaMA presented as an example of "GPTs" and [2] (CoT prompting) treated like a model (page 2); [2] is also mis-authored (Jason Wei vs. Jason Lee in references on page 9).
- Citation-content mismatch: Section 4.1 references [21] to support knowledge-graph screening of LLM outputs, but [21] concerns data-poisoning vulnerabilities, not knowledge-graph screening (pages 5 and 10).
- Reference formatting and selection issues: Malformed [11] (page 10); use of Swin Transformer [9] as the representative Transformer citation (page 2) is unconventional in this context.
- Lack of operationalization: The comparative framework and proposed strategies remain high-level without concrete taxonomies, algorithms, or evaluation protocols, especially for text-to-video claims.
- Overextension to text-to-video: Repeated mentions of text-to-video systems without specific literature coverage, method design, or empirical support tailored to that modality.

---

### Note · Reviewer_AIRevRelatedWork · 2025-10-06

**Related Work Check**

Please look at your references to confirm they are good.

**Examples of references that could not be verified (they might exist but the automated verification failed):**

- Reinforcement Learning: An Introduction by Richard S. Sutton, Andy Barto

---

### Decision · Program_Chairs · 2025-10-08

**Decision:**

Reject

**Comment:**

Thank you for submitting to Agents4Science 2025! We regret to inform you that your submission has not been accepted. Please see the reviews below for more information.